# Carvone Enantiomers Differentially Modulate IgE-Mediated Airway Inflammation in Mice

**DOI:** 10.3390/ijms21239209

**Published:** 2020-12-03

**Authors:** Jaime Ribeiro-Filho, Juliana da Silva Brandi, Hermann Ferreira Costa, Karina Carla de Paula Medeiros, Jacqueline Alves Leite, Damião Pergentino de Sousa, Márcia Regina Piuvezam

**Affiliations:** 1Laboratório de Investigação em Genética e Hematologia Translacional, Instituto Gonçalo Moniz, FIOCRUZ, Salvador 40296-710, Brazil; 2Departamento de Farmácia, Centro de Ciências da Saúde, Unifaminas Centro Universitário, Muriaé 36880-000, Brazil; julianabrandi20@hotmail.com; 3Faculdade de Medicina Nova Esperança, João Pessoa 58067-695, Brazil; hermanncosta@yahoo.com.br; 4Departamento de Morfologia, Centro de Biociências, UFRN, Natal 59072-970, Brazil; karinapm@yahoo.com; 5Departamento de Farmacologia, Instituto de Ciências Biológicas, UFG, Goiânia 74690-900, Brazil; jacquelineleite@ufg.br; 6Departamento de Ciências Farmacêuticas, Centro de Ciências da Saúde, UFPB, João Pessoa 58051-900, Brazil; damiao_desousa@yahoo.com.br; 7Laboratório de Imunofarmacologia, Departamento de Fisiologia e Patologia, UFPB, João Pessoa 58051-900, Brazil; mrpiuvezam@ltf.ufpb.br

**Keywords:** carvone, enantiomers, airway inflammation, eosinophils, anti-allergic, terpenes

## Abstract

Carvone is a monoterpene found in nature in the form of enantiomers (S- and R-). While previous research has demonstrated the anti-inflammatory and anti-allergic effects of carvone, the influence of carvone enantiomeric composition on its anti-allergic activity remains to be investigated. This study aimed to evaluate the anti-allergic activity of carvone enantiomers in a murine model of airway allergic inflammation induced by sensitization and challenge with ovalbumin (OVA). The oral treatment with R-carvone or S-carvone 1 h before each challenge inhibited the number of leukocytes and eosinophils in the bronchoalveolar lavage (BAL). R-carvone inhibited leukocyte infiltration and mucus production in the lung, which was correlated with decreased production of OVA-specific IgE in the serum and increased concentrations of IL-10 in the BAL. On the other hand, the administration of S-carvone had little inhibitory effect on inflammatory infiltration and mucus production in the lung, which might be associated with increased production of IFN-γ in the BAL. When administered 1 h before each sensitization, both enantiomers inhibited eosinophil recruitment to the BAL but failed in decreasing the titers of IgE in the serum of allergic mice. Our data indicate that carvone enantiomers differentially modulated IgE-mediated airway inflammation in mice. In conclusion, unlike S-carvone, R-carvone has the potential to be used in anti-allergic drug development.

## 1. Introduction

Asthma is one of the main non-communicable chronic inflammatory diseases. This condition affects people of all ages, with a prevalence of 10% in the adult population and even higher indices in children [1]. The complications of this disease can cause premature death and reduced quality of life, as well as demanding high treatment costs, making asthma a worldwide public health problem [2].

The pathophysiology of asthma is characterized by chronic inflammation of the respiratory tract that is mainly orchestrated by a Th2-type immune response. In allergic individuals, asthma is associated with increased IgE production, which has a crucial role in mast cell activation and early inflammatory reaction. The chronic inflammation is mainly characterized by eosinophil recruitment and activation in response to Th2 cell stimuli [3,4]. Increasing evidence has shown, however, the development of asthma in non-allergic individuals, where the pattern of the inflammatory response may vary significantly [5]. Asthmatic individuals may present several structural changes in the respiratory tract, such as enlarged smooth muscle, epithelial fibrosis, mucous cell metaplasia, increased vascularization and airway hyperreactivity. Asthma therapy is mainly done with inhaled corticosteroids and long-acting β_2_-agonists, but in some patients, muscarinic antagonists, phosphodiesterase inhibitors, oral corticosteroids, or anti-IgE drugs may be required. Nevertheless, asthma management remains challenging, and, therefore, the development of new therapies is urgent [6,7].

Essential oils are complex mixtures of volatile and aromatic compounds, among which terpenes stand out for their great chemical diversity and significant pharmacological effects [8]. Terpenes are formed by basic isoprene units, which may exist as hydrocarbons or present hydroxyl, ketones, carbonyl, or aldehyde groups, being named terpenoids [9].

Carvone (Figure 1) is a terpenoid ketone found in nature in the form of the enantiomers (S)-(+)-carvone (S-carvone) and (R)-(−)-carvone (R-carvone). S-carvone is the main constituent (50–70%) of the oil of *Carum carvi*, while R-carvone is an abundant constituent of mint oil (60–70%) obtained mainly from *Mentha spicata* [10]. This substance is industrially used as a flavoring agent [11] and germination inhibitor [12]. Previous studies have shown that carvone is a biologically active substance with antifungal [13] and anticancer [14] properties. Gonçalves and colleagues [15] demonstrated that this substance also has an analgesic action whose mechanism involves decreased peripheral nerve excitability.

Earlier reports have demonstrated that carvone treatment reduced neutrophil adherence and inhibited late-type hypersensitivity (DTH) reactions [16], suggesting that it has immunomodulatory properties. Accordingly, monoterpenes with a structure similar to that of carvone showed anti-inflammatory and anti-asthmatic activities [17,18]. However, the anti-inflammatory and anti-allergic effects of carvone remain to be investigated. Additionally, there are no studies reporting the influence of carvone enantiomeric composition with regard to its immunomodulatory properties.

Therefore, the objective of the present study is to evaluate the anti-allergic activity of carvone enantiomers in a murine model of allergic airway inflammation.

## 2. Results

### 2.1. Carvone Enantiomers Inhibit Eosinophil Recruitment in a Mice Model of Allergic Airway Inflammation

Ovalbumin-induced airway inflammation is characterized by an intense influx of eosinophils in the bronchoalveolar lavage (BAL) [19]. An allergic challenge with ovalbumin (OVA) (5%) in actively sensitized mice was observed to induce a significant increase in the number of total leukocytes in the BAL (Figure 2A). Differential counts identified eosinophils (Figure 2B) as the predominantly recruited leukocytes, which were found to be significantly increased in number in the untreated OVA-challenged group in comparison with the unchallenged control group. The allergic challenge induced a small (not significant) increase in the number of neutrophils (Figure 2D), while the number of mononuclear cells was not changed (Figure 2C). The oral treatment with both carvone enantiomers (10 mg/Kg) or dexamethasone (1 mg/Kg) 1 h before each OVA challenge caused a significant reduction in total leukocytes (Figure 2A) and eosinophil counts (Figure 2B) in comparison with the group of untreated OVA-challenged mice, thus demonstrating the inhibitory role played by these compounds concerning eosinophil recruitment during airway inflammation. The treatments with R-carvone or dexamethasone in OVA-challenged mice caused a mild increase in the number of mononuclear cells (Figure 2C), while neutrophil counts (Figure 2D) were found to be slightly increased in both OVA and S-carvone groups.

### 2.2. Effects of Carvone Enantiomers on Leukocyte Infiltration in the Lung of Allergic Mice

Following the analysis of leukocyte recruitment to the BAL, we analyzed the effects of carvone enantiomers on leukocyte infiltration into the lung tissue of BALB/c mice. To this end, these animals were treated and challenged as previously described, and 24 h after the last challenge, their lungs were removed and stained with hematoxylin-eosin (HE). As shown in Figure 3, the unchallenged animals (PBS) presented well-preserved parenchyma with no significant leukocyte infiltrate in the perivascular and perialveolar regions. In contrast, the lung of actively sensitized OVA-challenged mice (OVA) presented intense leukocyte infiltration in the peribronchiolar and perivascular areas, which was not observed in the pulmonary alveoli. The treatment with R-carvone (R-CAR) or dexamethasone (DEX) was shown to reduce the lung inflammatory infiltrates compared with the untreated OVA-challenged group. However, less evident inhibitory effect was observed with the treatment with S-carvone (S-CAR). An analysis of the inflammatory score in each group (Table 1) shows that most animals in the OVA group presented a severe inflammatory state, while the inflammatory state was significantly ameliorated by the treatment with R-carvone or dexamethasone. On the other hand, most animals in the S-carvone group presented a moderate inflammatory state, corroborating the histological aspects observed in Figure 3.

### 2.3. Effects of Carvone Enantiomers on Mucus Production in the Lung of Allergic Mice

Increased mucus production is a hallmark of allergic asthma [1]. Thus, histological sections were stained with periodic acid from Schiff (PAS) to identify the presence of mucopolysaccharides in the lung of allergic mice. Figure 4A shows that an allergic challenge in actively sensitized BALB/c mice induced an increase in mucus production compared with the unchallenged group. The treatment with (R)-(−)-carvone (R-CAR) or dexamethasone (DEX) reduced mucus production compared with the untreated OVA-challenged group (OVA), while, the treatment with (S)-(+)-carvone (S-CAR) had no beneficial effect on mucus production, providing evidence of a link between the differential inhibitory effects of carvone enantiomers on inflammatory cell infiltration and mucus production in the lung of allergic mice. This evidence was confirmed quantitatively using both the index (Figure 4B) and score (Table 2) of mucus production, corroborating the pharmacological phenomenon demonstrated in the representative photomicrographs.

### 2.4. Carvone Enantiomers Modulate IgE and Cytokine Production In Vivo

To assess the involvement adaptative immunity mechanisms on the anti-inflammatory properties of carvone enantiomers, we evaluated the production of IL-13, IL-10 and IFN-γ in the BAL, as well as the secretion of IgE in the serum of allergic mice. The treatment with S-carvone had no significant impact on the production of IL-13 (Figure 5A), IL-10 (Figure 5B) and IgE (Figure 5D). However, this monoterpene significantly increased the concentrations of IFN-γ (Figure 5C) in the BAL of allergic mice, which could be associated with the intense inflammatory infiltrate observed in the lungs, as well as with the increased neutrophil counts in the BAL of the mice treated with this compound. On the other hand, the group of mice treated with R-carvone presented significantly increased levels of IL-10, as well as lower titles of OVA-specific IgE. Nonetheless, the treatment with this enantiomer did not affect the production of IL-13 and IFN-γ compared to the untreated group. These data suggest that instead of modulating the allergic response through inhibition of Th2 cell-associated IL-13 production, R-carvone could be acting through stimulation of IL-10 synthesis and inhibition of IgE secretion. However, the mechanisms underlying R-carvone immunomodulatory effects remain to be further investigated.

### 2.5. Effects of the Treatment with Carvone Enantiomers on Allergic Sensitization 

As we demonstrated that the treatment with enantiomers 1 h before each challenge modulated airway inflammation in a murine model of asthma, this study evaluated the ability of these compounds to modulate allergic inflammation when the treatment was carried out before each sensitization. To this end, we analyzed two significant parameters of the OVA-triggered allergic inflammation: eosinophil recruitment and IgE production. Both enantiomers significantly reduced the number of eosinophils compared with the untreated OVA-challenged group (Figure 6A). However, none of these compounds reduced the titers of OVA-specific IgE (Figure 6B), indicating that treatment during the allergic challenge was crucial for the inhibitory effect demonstrated by R-carvone with regard to IgE production.

## 3. Discussion

Enantiomers are pairs of chiral molecules with one or more stereocenters. Despite having identical physical properties, enantiomers demonstrate different chemical behavior in chiral environments, which can affect their pharmacological properties. Accordingly, consistent evidence has shown that biological systems, including many endogenous drug receptors, membrane proteins and enzymes, are chiral compounds with unique three-dimensional configurations, whose activation might require some degree of stereoselectivity [20].

Accordingly, the administration of enantiomers in pharmacological models can result in at least one of the following responses: (1) the biological activity is attributed to only one of the enantiomers, while the other is inactive; (2) both enantiomers have identical or similar pharmacological properties under both qualitative and quantitative points of view; (3) the activity of the enantiomers is qualitatively equivalent but quantitatively different; and (4) the activities of the enantiomers are qualitatively different [21,22].

With regard to the pharmacokinetic properties, enantiomers can present significant differences in absorption, distribution, elimination and metabolism, since interaction with transporters, plasma proteins and metabolic enzymes may also involve some stereoselective discrimination [23,24]. Thus, considering the importance of molecular recognition in pharmacology and, therefore, the relevance of chirality for drug development [20], the present study evaluated the anti-allergic properties of R-carvone and S-carvone in a murine model of OVA-induced allergic asthma.

The accumulation of eosinophils in the airways is a characteristic feature of allergic asthma. Evidence has shown that these cells play critical roles in the pathogenesis of allergic diseases due to their ability to release both immunomodulatory and tissue-damaging mediators [25]. We demonstrated that the oral pre-treatment with R-carvone, S-carvone or dexamethasone significantly reduced the recruitment of total leukocytes and eosinophils to the BAL of allergic BALB/c mice, indicating that these treatments inhibited the eosinophilic inflammation triggered by the OVA challenge. However, the group of mice treated with S-carvone presented an increased number of neutrophils, while those animals undergoing R-carvone or dexamethasone treatment presented increased mononuclear cell counts. This finding suggests that carvone enantiomers differentially modulated the recruitment of leukocytes to the BAL of allergic mice. In this context, while neutrophil accumulation may be associated with an increased inflammatory state, the presence of mononuclear cells could indicate a change in the inflammatory state to a resolutive profile [26] in response to the treatments with R-carvone and dexamethasone, which is currently under investigation.

Studies have demonstrated that the recruitment and activation of eosinophils into the airways occurs in response to stimuli, such as eotaxin and IL-5, and contribute significantly to airway inflammation and hyperreactivity and, therefore, can be correlated with disease severity [27,28]. In fact, these cells produce a wide variety of inflammatory mediators, such as cis-leukotrienes, PAF, IL-3, IL-4, IL-5, IL-6, IL-8, IL-10, IL-13, RANTES, eotaxins, TGF-α, TGF-β and TNF-α, eosinophilic cationic protein (ECP), major basic protein (MBP) and eosinophilic peroxidase (EPO), which in addition to contributing to the asthma immunopathogenesis, can cause direct tissue damage [28]. 

The data of the present study confirmed that the reduction in the total number of leukocytes by the treatments was correlated with a direct effect on the recruitment of eosinophils, which represented approximately 70% of the cells identified in the BAL of the untreated allergic mice (not shown). To verify whether the effects of the treatment with carvone enantiomers on leukocyte infiltration into the BAL were also observed in the lung parenchyma of the mice, histological sections were observed under light microscopy. The lung of the mice treated with R-carvone presented reduced inflammatory infiltrates in the parenchyma, which was not observed in the S-carvone group, suggesting that the treatment with this enantiomer could affect the clearance of the cells through the lumen of the lung tissue. 

A study by Corry and colleagues, using a similar murine model of OVA-induced allergic asthma in matrix metalloproteinase 2 (MMP-2) knockout mice demonstrated that while these animals presented a significantly reduced number of leukocytes in the BAL, they showed increased accumulation of inflammatory cells into the lung parenchyma, indicating that MMP-2 plays a role in the removal of leukocytes from the airways of allergic mice [29]. Therefore, it is possible that S-carvone could be inhibiting the activity or expression of MMP-2 or other molecules involved in the elimination of inflammatory cells through the lung parenchyma. This hypothesis is corroborated by previous research showing that borneol, a monoterpene with a structure similar to that of carvone, reduced the activity of MMP-2 in the oral mucosa of BALB/c mice [30]. Nevertheless, further research is required to investigate the mechanisms associated with the accumulation of inflammatory cells in the lung of S-carvone-treated mice. In this context, our group is now dedicated to investigating the participation of different T cell populations, as well as the involvement of different MMP families on the airway inflammatory profile of allergic mice undergoing S-carvone treatment. 

Concerning the promising effects demonstrated by R-carvone on eosinophil recruitment in the BAL and inflammatory infiltration in the lung parenchyma, previous research has shown that this compound was found to induce the expression of FasL (CD95L) in vitro. This finding suggests that R-carvone could stimulate apoptosis in a wide variety of leukocytes expressing the Fas receptor (CD95), including eosinophils, neutrophils and dendritic cells [14]. However, considering that allergic inflammation results from a complex and multi-step immune cascade, R-carvone could be interfering with signaling cascades and or release/action of mediators with critical roles on eosinophil recruitment. While in this study we assess the involvement of some mediators potentially involved in this phenomenon, the investigation of cell death mechanism and signaling pathways modulated by R-carvone treatment will be targeted in the next step of our research.

Mucus hypersecretion by cells of the respiratory epithelium is an essential feature of allergic airway inflammation in severe asthma, resulting in the obstruction of the lower airways and progressive respiratory failure [31]. Therefore, following the evaluation of leukocyte infiltration, this work investigated the effect of carvone enantiomers on mucus production by analyzing histological sections stained with PAS. The treatment with R-carvone was found to cause a potent inhibition of mucus production. However, the treatment with S-carvone had no beneficial effects on mucus production, providing evidence of a link between the differential inhibitory effects of carvone enantiomers on inflammatory/eosinophilic inflammation and mucus production in the lung of allergic mice. This hypothesis is supported by evidence indicating that eosinophils can induce mucus secretion via the release of CysLTs [32]. Moreover, IL-13, a cytokine abundantly produced in allergic conditions, is an important stimulus for mucus production, as well as for many other pathophysiological characteristics of allergic airway inflammation [33] and, therefore, could be being differentially modulated by the action of carvone enantiomers.

To assess the involvement of adaptive immunity mechanisms on the anti-inflammatory properties of carvone enantiomers, we evaluated the production of IL-13, IL-10 and IFN-γ in the BAL, as well as the secretion of IgE in the serum of allergic mice. The oral administration of S-carvone significantly increased the concentrations of IFN-γ in the BAL of allergic the mice. On the other hand, R-carvone was found to stimulate IL-10 production and inhibit the production of OVA-specific IgE, suggesting an interference with mechanisms associated with the activation of regulatory T (Treg) cells and B lymphocytes, respectively. Since the R-carvone treatment exerted no inhibitory effect on IL-13 production, it is hypothesized the reduced mucus production observed in the lung of the mice treated with this monoterpene resulted from the inhibition of eosinophilic inflammation, although the interference of this terpene on eosinophil activation and mediator secretion needs to be further investigated.

IL-10 is a cytokine with key roles in immune regulation during asthma development. Evidence has suggested that, due to its immunosuppressive and anti-inflammatory properties, this cytokine plays a beneficial role in the pathogenesis of asthma, inhibiting cytokine production, eosinophilic inflammation, smooth muscle contraction and airway hyperreactivity [34]. Additionally, studies have demonstrated that CD4+ Th2 cells, which play critical roles in asthma pathogenesis, are directly regulated by IL-10 during allergic airway inflammation [35]. Therefore, the increased IL-10 concentrations found in the BAL of R-carvone-treated mice could justify, at least partially, the reduction in airway inflammatory parameters observed in these animals. In addition, studies with biological agents have demonstrated that inhibition of IgE in patients with moderate and severe asthma resulted in improvement in patient symptoms and quality of life [36], and therefore the inhibition of IgE production, even though not very expressive, may have contributed to the anti-allergic effects of R-carvone observed in our study.

However, since the effect of R-carvone on IL-10 production was noticeably more evident than its inhibitory effect on IgE production, it is hypothesized that the actions of this monoterpene are significantly associated with the modulation of the inflammatory profile in the pulmonary environment, which corroborates the leukocyte recruitment data showing a decrease in the proportion of eosinophils and an increase in the proportion of mononuclear cells, suggesting that R-carvone may have immunoregulatory and pro-resolving effects. This hypothesis will be investigated using a murine model of resolution of allergic airway inflammation [37] to evaluate the effects of the treatment with carvone on eosinophil apoptosis, migration of resolutive macrophages and production of pro-resolving mediators such as resolvins and lipoxins. Although the present study has not characterized the mechanisms associated with increased IL-10 production in R-carvone-treated mice, evidence has identified alveolar macrophages and regulatory T cells as important cellular sources of this cytokine [35] and, therefore, the investigation of the effect of this terpene on these cells will contribute significantly to the elucidation of their immunoregulatory mechanisms.

The results of the present study also demonstrated a differential action of carvone enantiomers on the production of IFN-γ, whose role in IgE-dependent allergic inflammation has been previously reported [38]. Studies indicate that increased production of IFN-γ after the establishment of a Th2 response contributes to the increase of the inflammatory process, possibly by inducing the activation of epithelial cells and the contraction of smooth muscle cells and by promoting airway remodeling [39,40]. Here, we demonstrated that the treatment with S-carvone resulted in increased IFN-γ production, which can be associated with the inflammatory state of the mice, as demonstrated by the lung inflammation and mucus production score values, as well as by the increased neutrophil counts in the BAL. This hypothesis is supported by evidence showing neutrophils as important sources of IFN-γ in both humans and mice [41,42]. Additionally, studies have demonstrated that type 1 responses and IFN-γ play critical roles in neutrophilic airway inflammation, nitric oxide production and poor response to corticosteroids, contributing to the disease severity [43]. Nevertheless, a more detailed analysis is required to characterize the populations of IFN-γ-producing cells in the BAL of allergic mice treated with S-carvone.

To evaluate the effectiveness of carvone enantiomers in modulating airway inflammation by interference on allergic sensitization, we analyzed eosinophil recruitment and IgE production in mice treated before each sensitization, instead of during the allergic challenge. While both enantiomers were found to significantly reduce eosinophil infiltration in the BAL of allergic mice, none of these compounds reduced the titers of OVA-specific IgE. This finding suggests that the R-carvone-mediated inhibition of IgE production does not occur due to a direct action on B cell function, but through immunomodulatory mechanisms that indirectly affect IgE production during the development of airway inflammation in response to the OVA challenge. The possibility that R-carvone may directly interfere with mast cell activation in response to the OVA-challenge cannot be ruled out. This evidence is in line with the observation that this monoterpene caused little expressive (24.8%) decrease in the IgE titers, comparatively with the inhibition of other inflammatory parameters in mice undergoing oral treatment before each allergic challenge. Nonetheless, additional studies are necessary to evaluate the effectiveness of R-carvone treatment at different stages of the allergic cascade triggered by sensitization and challenge with OVA, in particular, the effects of the treatment with R-carvone before each sensitization on allergic parameters such as leukocyte infiltration, mucus secretion, inflammatory mediator production and airway hyperresponsiveness. Besides, we intend to investigate the effect of the post-treatment (after the allergic challenge) with this compound, since it more accurately simulates the treatment in asthmatic individuals.

Importantly, it remains unknown how the treatment with these enantiomers before each sensitization reduced the number of eosinophils without changing the levels of IgE. Studies indicate that carvone metabolism in humans occurs around 2.4 h. However, the pharmacokinetic profile of carvone enantiomers in mice remains to be investigated. Nevertheless, evidence has suggested that carvone metabolism is likely to be different in humans and rats, as due to enterohepatic circulation, rats can be more sensitive than humans to terpenes [44]. Therefore, studies are needed to elucidate the half-life or carvone enantiomers in mice, as well as the existence of active metabolites and their immunomodulatory effects in the long-term, including regarding the modulation of IgE production during the allergic challenge.

The effects of monoterpenes on allergic asthma remain poorly characterized. Nevertheless, studies have shown that 1,8-cineole (eucalyptol) presented anti-inflammatory and mucolytic effects in a murine model of asthma, by inhibiting the production of cytokines such as TNF-α, IL-1β, IL-4 and IL-5 by lymphocytes, as well as by inhibiting the secretion of TNF-α, IL-1β, IL-6 and IL-8 by peripheral blood mononuclear cells [45]. On the other hand, the anti-asthmatic activity of L-menthol has been attributed, at least in part, to the inhibition of lipid mediator production [46]. To date, few studies have investigated the impact of chirality on the activity of drugs acting on immune diseases. Nonetheless, some evidence suggests that chirality significantly affects T cell-dependent responses [47]. Accordingly, Hong and collaborators [48] reported that changes in the chemical structure and chirality of compound naphthopyran affected its action in regulating immune responses mediated by helper T cells. A comparative study of (R)- and (S)-albuterol in a murine model of asthma revealed that although both compounds were able to reduce leukocyte infiltration and mucus production in the lung of allergic mice, only (R)-albuterol was found to inhibit the production of IL-4. Additionally, it was demonstrated that (S)-albuterol increased allergen-induced interstitial edema [49]. 

These findings indicate that differences in the pharmacological responses of enantiomers may occur due to differences in the three-dimensional arrangement of the atoms in these molecules. In this context, although both carvone enantiomers were found to inhibit eosinophil recruitment to the BAL, only R-carvone presented anti-allergic effects that encourage its use in anti-asthmatic drug development, as attested by its inhibitory effects on OVA-specific IgE production, leukocyte infiltration and mucus secretion in the lung of allergic mice. However, additional studies are required to clarify the mechanisms associated with leukocyte accumulation, as well as its impacts on airway hyperresponsiveness in S-carvone-treated mice.

The OVA-induced airway allergic inflammation in mice is a well-established experimental model to investigate the effectiveness of new drug candidates for the treatments of allergic asthma [7]. However, recent research by Wu and colleagues [50] raised important issues related to inhibition of IgE production by PGE_2_ in OVA-induced asthma. Therefore, although we believe that this model is appropriate to demonstrate the differential effects of carvone enantiomers on IgE-mediated allergic inflammation, experiments using human-relevant allergens like house dust mite should be conducted to better characterize the therapeutic potential of R-carvone as an anti-asthmatic compound. 

In conclusion, carvone enantiomers differentially modulated IgE-mediated airway inflammation in mice. While R-carvone presented promising anti-asthmatic potential, the use of S-carvone failed in inhibited most inflammatory parameters in addition to increasing the concentrations of IFN-γ in the BAL of allergic mice, which could contribute to increased airway inflammation. Of note, this is the first study comparing the anti-allergic activity of the carvone enantiomers, and, therefore, further research is required to fully characterize the anti-allergic properties and molecular mechanisms underlying the anti-inflammatory and immunomodulatory action of R-carvone in allergic asthma models. 

## 4. Materials and Methods 

### 4.1. Drugs

S-carvone and R-carvone were purchased from Sigma Aldrich (SIGMA-ALDRICH, St. Louis, MO, USA). Both substances were emulsified in 5% tween 80 (VETEC, Rio de Janeiro, Brazil), diluted in PBS and administered orally by single gavage at 10 mg/Kg. Briefly, each enantiomer emulsion was prepared by weighing 3 mg of the compound in test tubes. The substances were emulsified in with 0.15 mL of tween 80 and after homogenization, the volume was adjusted to 3 mL with PBS. Each animal received 0.1 mL of this emulsion per 10 g of body weight, corresponding to a dose of 10 mg/Kg. Equal volume of dexamethasone (1 mg/Kg) was administered orally by single gavage as a control drug. Ovalbumin (OVA, grade V) was purchased from SIGMA Chemical (St. Louis, MO, USA) and used in both sensitization and challenge protocols. Al(OH)_3_ and Evans Blue were purchased from VETEC (Rio de Janeiro, RJ, USA). PBS containing 5% tween 80 (vehicle) was administered orally to the mice in both the control and OVA groups.

### 4.2. Animals

Female BALB/c isogenic mice aged 6 to 8 weeks and weighing between 20 g and 25 g and Wistar rats (*Rattus norvegicus*) weighing about 200 g were supplied by the Biotério Prof. Thomas George of the Pharmaceutical Technology Laboratory Delby Fernandes de Medeiros (LTF) of the Federal University of Paraiba (UFPB). The animals were subjected to a 12-h light and dark cycle, maintained at a temperature of 25 ± 2 °C with free access to water and balanced pellet feed (Ralston Purina Co., St. Louis, MO, USA). The experiments were conducted in order to preserve animal welfare, according to the animal care guide [51]. Euthanasia was performed by anesthetic overdose (100 mg/Kg of ketamine and 14 mg/Kg of xylazine Virbac, São Paulo, SP, Brazil). All protocols of this study were approved by the Animal Research Ethics Committee (CEPA) of the LTF of the Federal University of Paraíba on 6 April 2008 (Opinion No. 0604/08).

### 4.3. Airway Allergic Inflammation in Actively Sensitized Mice

Airway inflammation was induced in actively sensitized mice, as described by Lloyd et al. [52]. Briefly, female BALB/c mice (*n* = 6–8) were sensitized intraperitoneally (i.p) with 200 μL of OVA (10 μg) emulsified with Al(OH)_3_ (10 mg/mL) in PBS. From day 19 to day 24 after sensitization, the mice were challenged daily for 20 min with OVA (5%) in PBS by aerosol. A group of mice receiving aerosolized PBS was used as the negative control. The challenge was performed in a 30 × 20 × 10 cm acrylic chamber connected to an ultrasonic nebulizer. Twenty-four hours after the last challenge, the animals were euthanized as described before, and the trachea was surgically exposed. The bronchoalveolar lavage (BAL) was collected by washing the lungs with 1 mL of HBSS. 

### 4.4. Leukocyte Counting

Samples of the bronchoalveolar lavages were diluted (1:4) in Turk fluid (2% acetic acid), and then, total leukocytes were counted using a Neubauer chamber under light microscopy. Differential counts were performed by analyzing the cell morphology under an objective lens at 100X magnifications. Cytospins stained by the May–Grunwald–Giemsa method were used to determine the number of eosinophils, neutrophils and mononuclear cells. 

### 4.5. Lung Histology

Female BALB/c mice (n = 6–8) were sensitized, treated and challenged, as described in the airway allergic inflammation protocol above. Twenty-four hours after the last challenge, the animals were euthanized as described above, the thorax was surgically exposed and the lungs were washed with PBS (20 mL) by cardiac perfusion. The lungs were then removed, fixed in 10% formaldehyde for 24 h and then fixed in increasing concentrations of ethanol (60%, 70%, 80%, 90% and 100%) and embedded in paraffin. Lung sections of 5-μm thickness were stained with hematoxylin-eosin (HE) and periodic acid-Schiff (PAS) according to standard protocols for analysis of leukocyte infiltrate and mucus secretion, respectively. Additionally, HE and PAS-stained slides were used for the evaluation of OVA-induced histological changes associated to inflammatory cell infiltration and mucus production. Histological changes were scored as follows: (0) not detectable, (1) mild, (2) moderate and (3) severe. The histopathological examinations were performed by a blind pathologist through bright-field microscopy. For morphometric analysis, the images of PAS-stained slides were acquired using an OLYMPUS CX21 optical microscope (OLYMPUS, Center Valley, PA, USA) coded for blinded analysis. The circumference of the PAS-stained area was electronically measured using the ImageJ software (NIH, Bethesda, MD, USA) and the mucus index was determined and expressed as percentage of the total bronchial area [53]. The photomicrographs of the slides stained with HE and PAS were photographed at 400× and 100× magnifications, respectively. These analyses were performed with 5 animals per group.

### 4.6. Quantification of OVA-Specific IgE by Passive Cutaneous Anaphylaxis (PCA)

Twenty-four hours after the last challenge in the allergic airway inflammation protocol, the mice were anesthetized as previously described and 50 µL of blood was collected from mice by an orbital puncture. Blood samples were centrifuged at 6000× *g* for 10 min at 4 °C, and the OVA-specific IgE titers in the serum were determined using the passive cutaneous anaphylaxis (PCA) reaction, as described by Holt et al. [54]. Briefly, serum dilutions (ranging from 1:8 to 1: 1024) were inoculated intradermally on the backs of Wistar rats, and 48 h later, each animal was challenged intravenously (i.v.) with 500 µL of a solution containing OVA (2 mg) dissolved 1% Evans Blue. After 30 min, the rats were euthanized, and OVA-specific IgE titers were measured. The highest serum dilution resulting in a 5-mm diameter bluish spot was defined as the PCA titer [55].

### 4.7. Cytokine Quantification

Samples of the BAL were centrifuged at 500× *g* for 8 min at 4 °C to obtain the supernatant. The concentrations of IFN-γ, IL-13 and L-10 were measured by ELISA, using DuoSet kits according to the manufacturer’s instructions (BIOSCIENCE, San Diego, CA, USA). Briefly, 96-well plates were sensitized with the capture antibody, incubated at 4 °C for 18 h, washed with PBS-Tween 20 (0.05%) (SIGMA-ALDRICH, St. Louis, MO, USA) and incubated with blocking buffer (PBS in 10% bovine fetal serum) for 1 h. Following this step, the plates were incubated with the recombinant antibody at 4 °C for 18 h and then incubated with the detection antibody at 4 °C for 18 h. After this period, the plates were incubated with avidin-peroxidase for 30 min at room temperature and filled with the substrate (TMB + H_2_O_2_) to react for 15 min. The reaction was stopped by adding 1N sulfuric acid, and the readings were performed in a spectrophotometer at 450 nm. The concentrations of the cytokines in the BAL were calculated from standard curves, with the following assay ranges: IFN-γ: 15.1–2000 pg/mL, IL-13: 3.78–500 pg/mL and IL-10: 31.25–4000 pg/mL.

### 4.8. Statistical Analyses

Data were analyzed by one-way ANOVA followed by Tukey’s post-test using GraphPad Prism software version 5.02 (GraphPad, San Diego, CA, USA, 2016). Values are expressed as means ± Standard Error of Mean (SEM) and are representative of two independent experiments. Statistical significance was considered when *p* < 0.05.

## Figures and Tables

**Figure 1 ijms-21-09209-f001:**
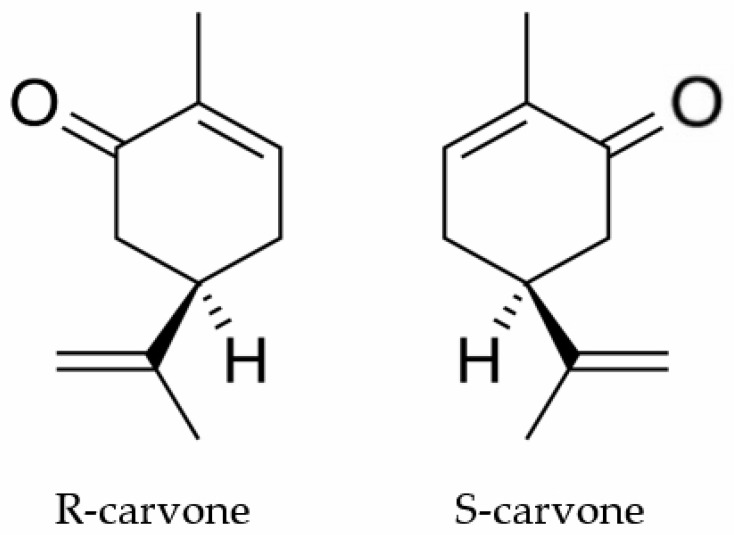
The chemical structures of R-carvone and S-carvone.

**Figure 2 ijms-21-09209-f002:**
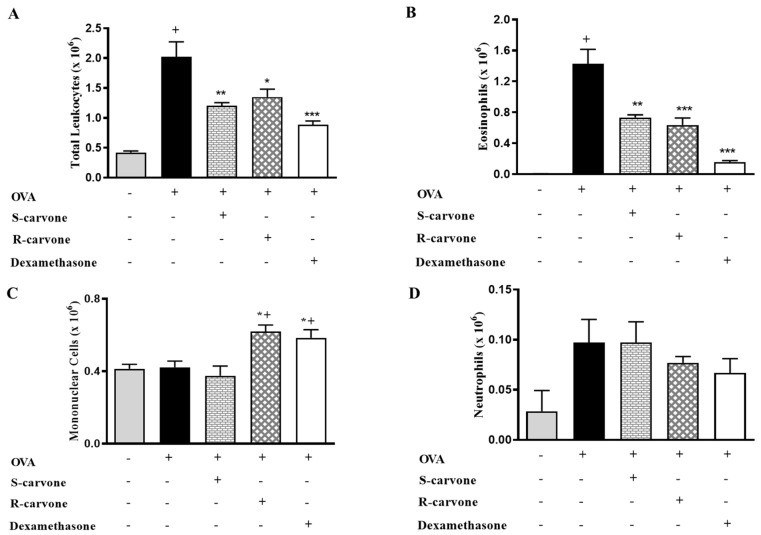
Effects of carvone enantiomers on leukocyte recruitment in allergic airway inflammation. (**A**) Total leukocytes, (**B**) Eosinophils, (**C**) Mononuclear cells and (**D**) Neutrophils per bronchoalveolar lavage (BAL) of BALB/c mice orally pre-treated with R-carvone (10 mg/Kg), S-carvone (10 mg/Kg) or dexamethasone (1 mg/Kg), counted under light microscopy 24 h after the last ovalbumin (OVA)-challenge. Results are expressed as means ± SEM and are representative of two independent experiments performed with 6–8 animals per group. + significant difference (*p* < 0.05) from the unchallenged group; * (*p* < 0.05), ** (*p* < 0.01) and *** (*p* < 0.001) express significant difference from the untreated OVA-challenged group. Statistical significance was determined with one-way ANOVA and post hoc Tukey test.

**Figure 3 ijms-21-09209-f003:**
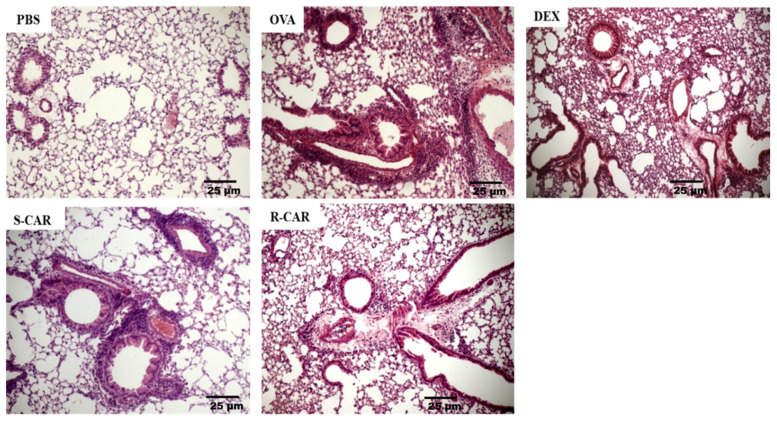
Effects of carvone enantiomers on leukocyte infiltration in the lung of allergic mice following OVA-induced allergic airway inflammation. Twenty-four hours after the last challenge, lungs were fixed in 10% formalin, embedded in paraffin and sectioned. The tissue section slides were stained with HE. OVA-sensitized mice were treated with R-carvone (10 mg/Kg), S-carvone (10 mg/Kg) or dexamethasone (1 mg/Kg) or saline (vehicle) 1 h prior to each challenge. These analyses were performed by a blind observer using a light microscope with 400× magnifications. Each photomicrograph is representative of two independent experiments performed with five animals per group.

**Figure 4 ijms-21-09209-f004:**
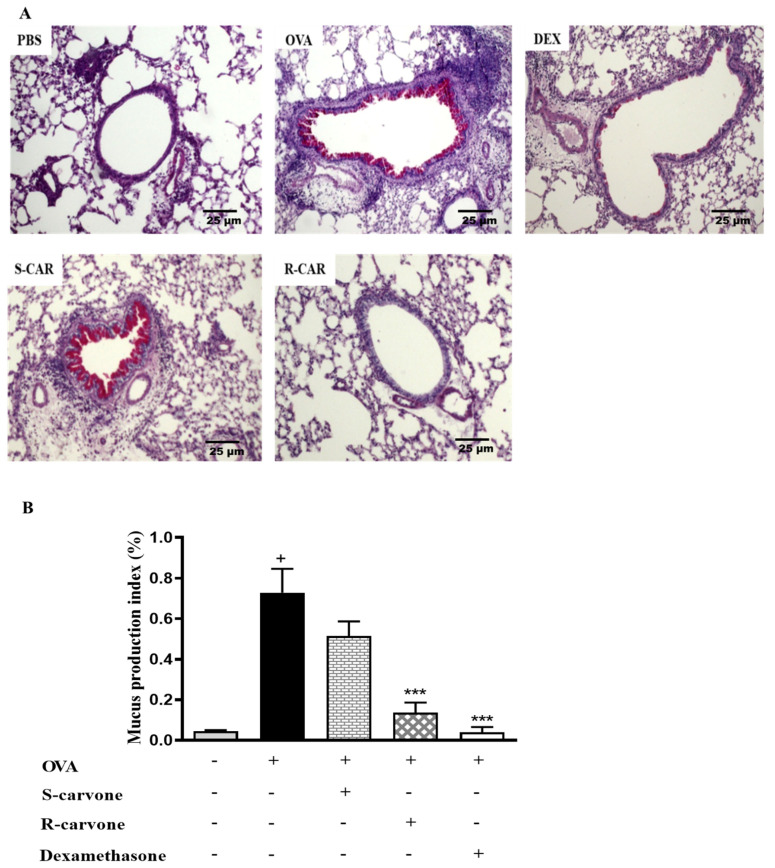
Effects of carvone enantiomers on mucus production in the lung of allergic mice following OVA-induced allergic airway inflammation. (**A**) Photomicrographs of lungs stained with periodic acid from Schiff (PAS). (**B**) The mucus production index on PAS-stained slides was determined using the Image J software and expressed as means ± SEM from five animals. + significant difference (*p* < 0.05) from the unchallenged group; *** (*p* < 0.001) express significant difference from the untreated OVA-challenged group. Statistical significance was determined with one-way ANOVA and post hoc Tukey test. Twenty-four hours after the last challenge, lungs were fixed in 10% formalin, embedded in paraffin and sectioned. The tissue section slides were stained with PAS. OVA-sensitized mice were treated with R-carvone (10 mg/Kg), S-carvone (10 mg/Kg), dexamethasone (1 mg/Kg) or PBS (vehicle) 1 h prior to each challenge. These analyses were performed by a blind observer using a light microscope with 100× magnifications. Each photomicrograph is representative of two independent experiments performed with five animals per group.

**Figure 5 ijms-21-09209-f005:**
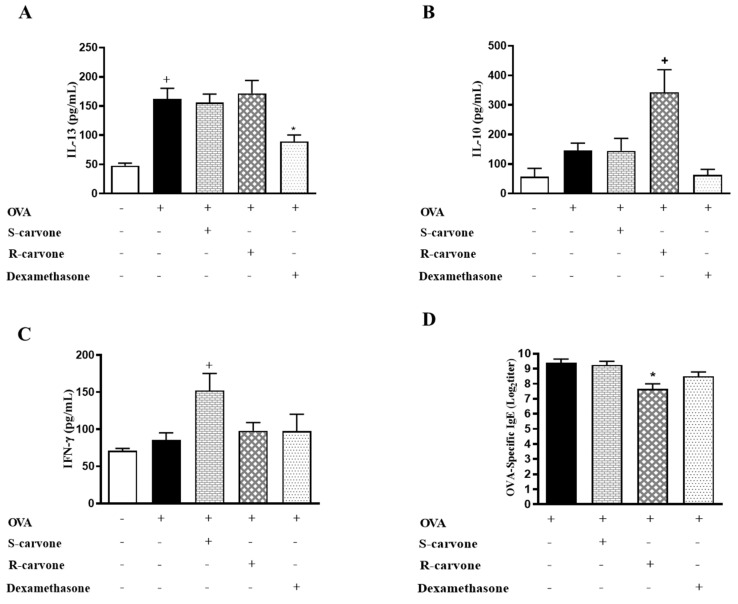
Effect of carvone enantiomers on cytokine and IgE production in BALB/c mice orally pre-treated with R-carvone (10 mg/Kg) and S-carvone (10 mg/Kg). Concentrations of IL-13 (**A**), IL-10 (**B**), IFN-γ (**C**) in the BAL were determined by ELISA. Titers of OVA-specific IgE in the serum were assessed using the PCA test (**D**). All analyses were performed 24 h after the last OVA-challenge. These results are expressed as the mean ± SEM and are representative of two independent experiments performed with 6–8 animals per group. + significant difference (*p* < 0.05) from the unchallenged group; * significant difference (*p* < 0.05) from the untreated OVA-challenged group. Statistical significance was determined with one-way ANOVA and post hoc Tukey test. Assay ranges—IFN-γ: 15.1–2000 pg/mL, IL-13: 3.78–500 pg/mL and IL-10: 31.25–4000 pg/mL.

**Figure 6 ijms-21-09209-f006:**
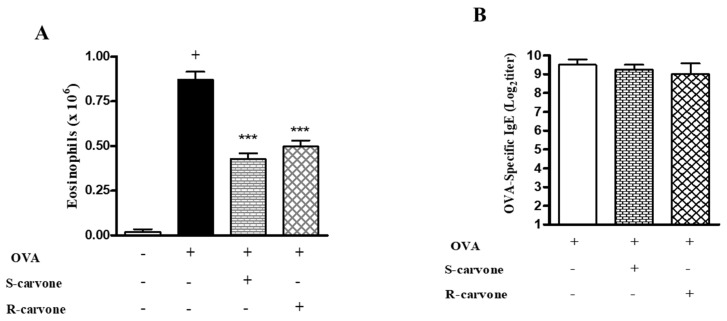
Effects of the treatment with carvone enantiomers on allergic sensitization. BALB/c mice were orally pre-treated with R-carvone (10 mg/Kg) or S-carvone (10 mg/Kg) 1 h before each sensitization. (**A**) Eosinophils per BAL counted under light microscopy and (**B**) Titers of OVA-specific IgE in the serum were assessed using the PCA test 24 h after the last OVA-challenge. Results are expressed as means ± SEM and are representative of two independent experiments performed with 6–8 animals per group. + significant difference (*p* < 0.05) from the unchallenged group; *** significant difference (*p* < 0.001) from the untreated OVA-challenged group. Statistical significance was determined with one-way ANOVA and post hoc Tukey test.

**Table 1 ijms-21-09209-t001:** Lung inflammation score per experimental group (*n* = 5).

Group	Not Detectable (0)	Mild (1)	Moderate (2)	Severe (3)
PBS	4	1	-	-
OVA	-	-	2	3
S-carvone	-	1	4	-
R-carvone	1	2	2	-
Dexamethasone	1	4	-	-

**Table 2 ijms-21-09209-t002:** Mucus production score per experimental group (*n* = 5).

Group	Not Detectable (0)	Mild (1)	Moderate (2)	Severe (3)
PBS	-	-	-	-
OVA	-	2	3	-
S-carvone	-	1	3	1
R-carvone	1	2	2	-
Dexamethasone	1	4	-	-

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
