# Peer review of "Carvone Enantiomers Differentially Modulate IgE-Mediated Airway Inflammation in Mice"

_ijms, 2020, doi:10.3390/ijms21239209_

Round 1

Reviewer 1 Report

The article has been much improved, therefore I don't have any further comments and I recommend it to accept in the present from.

Reviewer 2 Report

In this pilot study the anti-inflammatory properties of s- and r-carvones were analyzed and compared in a murine model of allergic airway disease.

But of lung function the impact of the treatment of asthma hallmarks like eosinophilia in BAL, inflammation in lung tissue and goblet cell metaplasia was analyzed.

The authors give a first hint on the immunomodulatory function of the carvones. The authors answered all my concerns regarding the present form of the manuscript.

However, further studies are needed to substantiate the mode of action of the molecules.

This manuscript is a resubmission of an earlier submission. The following is a list of the peer review reports and author responses from that submission.

Round 1

Reviewer 1 Report

Carvones are monoterpernes, which show anti-inflammatory properties. They are found in nature as S- or R- configurated enantiomers. Aim of the present study is to demonstrate the influence of the configuration of the Carvones on their anti-inflammatory properties.

Using a murine chicken Ovalbumin (OVA) specific allergic asthma model impact of treatment with R- or S-Carvones during sensitization or challenge was investigated.

Animals were sensitized to Ovalbumin (Day 1 and 10). To provoke the allergic airway disease animals were challenged with nebulized OVA on the days 19-24. To analyze the impact of the different carvone enantiomers S- or R-Carvones were orally administered either before each challenge or before each sensitization step.

To characterize the influence on the allergic airway disease eosinophils in BAL, inflammatory influx in lung tissue and development of a goblet cell metaplasia was investigated. Additionally concentration of IL-13, IL-10 and IFNy in BAL and OVA-specific IgE was measured.

The group could demonstrate that both S- and R-carvones given during challenge or sensitization phase are able to reduce the numbers of BAL eosinophils. However, only R-Carvones attenuate lung tissue infiltrating leukocytes and induction of mucus secreting goblet cells.

Moreover, administration of R-carvones increased IL-10 concentrations in BAL and reduced titers of OVA specific IgE in sera. Surprisingly, both effects were not observable upon application of S-carvones, on the contrary, the S-conformation led to increased concentration of IFN-y.

The group concludes that R- but not S-conformations of Carvones have beneficial anti-allergic properties in a murine asthma model.

The paper is well written. However, several information are missing and the observations are almost strictly descriptive.

Major comments:

  1. The analysis and presentation of the asthma phenotype:
    1. The chosen model sounds solid, unfortunately in M&M some information are missing:
      • Which OVA was used?
      • Was it emulsified with Alum.
      • Which concentration and amount was administered.
      • Please check citation: Line 364: 45 seems not to be the Lloyd Paper.
    1. Please enhance asthma readout information:
      • Cellular composition of BAL: Here only total numbers of eosinophils are shown. To receive a better overview on possible immune modulatory effects of the Carvones and the asthma model per se it would be recommendable to present different numbers of macrophages, lymphocytes, neutrophils and eosinophils.

                       Especially with regard to the fact that increased    concentrations of IFNy a Th1 cytokine associated with neutrophil responses in asthma.

      • Please add a scoring of the inflammatory state: A scoring of inflammatory influx of lung slides of all analyzed animals would help to better estimate the effectiveness of the Carvone treatment as a single picture does.
      • Likewise, please score mucus production. A score like number of PAS positive cells associated with length of basal membrane would help to estimate the treatment effect.
      • Lung functions: the analysis of lung function is completely missing.
    1. OVA specific Immunoglobulines: Please explain why a rat specific passive cutaneous anaphylaxis model instead of an immunological assay like ELISA. Presentation of OVA specific IgG1 and IgG2b concentration would help to understand if Carvones modulate Th1/Th2 and B cell responses.
    1. Nowadays, more and more models using human relevant allergens, like HDM, are used. Possible downsides of an OVA model should be at least mentioned in the discussion.
  1. Immunomodulatory effects: The group concludes that Carvones affect the adaptive immune response via induction of regulatory IL-10 producing cells (R-Carvone) or a shift to a Th1 response (S-Carvone). Cytokine measurements in BAL were the only experiments performed to substantiate the hypothesis of effect mechanism of the Carvones. Further experiments are mandatory to prove the effect mechanism in general and later on in detail.
    1. Are number of Tregs different in lung and draining lymph node.
    2. Are there differences in the local (lung/draining lymph node) response vs the systemic immune response (spleen).

                Composition of immune cells.

                Cytokine production upon allergen resimulation.

  1. The group observes a reduction of eosinophils in BAL upon treatment with Carvones during challenge and sensitization phase. Surprisingly, a reduction of immunoglobulins was only observable following treatment during challenge phase.
    1. How do you explain the anti-inflammatory effectiveness of treatment during both phases.
    2. Were lung tissue inflammation and mucus production also suppressed following treatment in the sensitization phase.
    3. Since, systemic antibodies are induced during sensitization phase, how do you explain the effectiveness of Carvones only during treatment in the challenge phase.
  2. A little bit more information of the S- and R-Carvones from Sigma should be provided.
    1. How are the preparations of both Carvones performed, are they similar?
    2. The Carvones are diluted in 5% Tween 80 and further diluted in PBS. Are the dilutions comparable? Was diluted Tween 80 tested as a control?
  3. In the discussion a differentiated regulation of MMP-2 was presented as an explanation why eosinophils were reduced upon both treatments in BAL but only R-Carvones led to a reduction of inflammatory cells in lung tissue.
    1. A demonstration of a differential regulation of MMP-2 would be beneficial for the manuscript.

Minor comments

  1. Please check citations, some citations are missing (46 and further), some citations seems to be wrong.
  2. Please indicate magnification of histological pictures (Fig.2 and Fig.3)
  3. As mentioned before, please ad grades, concentrations and application amounts OVA and all Carvone treatments.

Author Response

Point 1: The analysis and presentation of the asthma phenotype:

The chosen model sounds solid, unfortunately in M&M some information are missing:

Which OVA was used?

Was it emulsified with Alum?

Which concentration and amount was administered?

Response 1: Thank you for your contribution. We used OVA grade V emulsified with Alum for sensitization. The challenge was performed using OVA (5%) in PBS by aerosol for 20 min. This information has been in included in sections 4.1 and 4.3, as follows:

Lines 427-428: “…Ovalbumin (OVA, grade V) was purchased from SIGMA Chemical (St. Louis, MO, USA) and used in both sensitization and challenge protocols.”

Lines 445-447: “…Briefly, female BALB/c mice (n= 6-8) were sensitized intraperitoneally (i.p) with 200 μL of OVA (10 μg) emulsified with Al(OH)3 (10 mg/mL) in PBS. From day 19 to day 24 after sensitization, the mice were challenged daily for 20 min with OVA (5%) in PBS by aerosol.”

Point 2: Please check citation: Line 364: 45 seems not to be the Lloyd Paper

Response 2: Thank you. The reference order was incorrect. We have carefully reviewed the references in our revised manuscript.

Point 3: Please enhance asthma readout information:

Cellular composition of BAL: Here only total numbers of eosinophils are shown. To receive a better overview of possible immune modulatory effects of the Carvones and the asthma model per se it would be recommendable to present different numbers of macrophages, lymphocytes, neutrophils and eosinophils.

Especially with regard to the fact that increased    concentrations of IFNy a Th1 cytokine associated with neutrophil responses in asthma.

Response 3: Thank you for your suggestion. In our revised manuscript we provide a more detailed cellular composition of BAL. The number of total leukocytes (Fig 2A), eosinophils (Fig. 2B), mononuclear cells (Fig. 2C), and neutrophils (Fig. 2D) were included. Unfortunately, in our analysis, we did not distinguish the types of mononuclear cells. However, soon, we intend to carry out new experiments to evaluate both macrophage subtypes as well as the different lymphocyte subpopulations, as well as their role in the production of mediators in this model.

The description of these new data was discussed in materials and methods, results, and discussion.

With regard to the correlation between the leukocyte counts and IFN-γ production, we have made significant efforts to improve our discussion. References 40-43 were added to the manuscript to support our statements, as follows:

Lines 344-352: “Here, we demonstrated that the treatment with S-carvone resulted in increased IFN-γ production, which can be associated with the inflammatory state of the mice, as demonstrated by the lung inflammation and mucus production score values, as well as by the increased neutrophil counts in the BAL. This hypothesis is supported by evidence showing neutrophils as important sources of IFN-γ in both humans and mice [40,41]. Additionally, studies have demonstrated that type 1 responses and IFN-γ play critical roles in neutrophilic airway inflammation, nitric oxide production, and poor response to corticosteroids, contributing to the disease severity [42]. Nevertheless, a more detailed analysis is required to characterize the populations of IFN-γ-producing cells in the BAL of allergic mice treated with S-carvone.”

Point 4: Please add a scoring of the inflammatory state:

A scoring of the inflammatory influx of lung slides of all analyzed animals would help to better estimate the effectiveness of the Carvone treatment as a single picture does.

Likewise, please score mucus production. A score like number of PAS positive cells associated with length of basal membrane would help to estimate the treatment effect.

Response 4: Thank you for your suggestion. We totally agree that scoring of inflammatory influx and mucus production improves the estimation of the effectiveness of the Carvone treatment. Despite limited access to research laboratories due to the severity of the COVID-19 pandemic in Brazil, we analyzed the slides and made inflammatory and mucus production scores. These data were inserted in the results (Table 1, Table 2, and figure 4B) and its repercussion for the interpretation of the effects of the enantiomers was discussed in the revised manuscript.

Point 5: Lung functions: the analysis of lung function is completely missing.

OVA specific Immunoglobulines: Please explain why a rat specific passive cutaneous anaphylaxis model instead of an immunological assay like ELISA. Presentation of OVA specific IgG1 and IgG2b concentration would help to understand if Carvones modulate Th1/Th2 and B cell responses.

Response 5: Thank you for mentioning the importance of determining IgG1 and IgG2 to analyze the balance between Th1/Th2 profiles and, we agreed with it. However, in our manuscript, we chose to measure the OVA-specific IgE in a PCA reaction due to its high specificity and widely used to evaluate the effectiveness of antiallergic compounds to inhibit OVA-specific IgE production. Please, see some references below. In fact, R-carvone presented an antiallergic activity associated with the inhibition of eosinophil recruitment, decreased mucus production, and an increase of IgE-dependent IL-10 production, which strongly suggest that this enantiomer modulates Th2 responses in mice. However, as this enantiomer failed in decreasing IL-13 and showed mild effects on IgE production, its effects on T cell activation remains to be further investigated.

PCA references:

  1. Costa, H.F.; Bezerra-Santos, C.R.; Barbosa Filho, J.M.; Martins, M.A.; Piuvezam, M. R. Warifteine, a bisbenzylisoquinoline alkaloid, decreases immediate allergic and thermal hyperalgesic reactions in sensitized animals. Int Immunopharmacol 2008, 8(4), 519-525.
  2. Holt, P.G.; Rose, A.H.; Batty, J.E.; Turner, K. J. Induction of adjuvant independent IgE responses in inbred mice: primary, secondary, and persistent IgE responses to ovalbumin and ovomucoid. Int Arch Allergy Immunol 1981, 65(1), 42-50.
  3. Xia XC, Chen Q, Liu K, et al. Mycoepoxydiene inhibits antigen-stimulated activation of mast cells and suppresses IgE-mediated anaphylaxis in mice. Int Immunopharmacol. 2013;17(2):336-341. doi:10.1016/j.intimp.2013.06.02

Point 6: Nowadays, more and more models using human relevant allergens, like HDM, are used. Possible downsides of an OVA model should be at least mentioned in the discussion.

Response 6: The OVA-induced pulmonary allergic inflammation in mice is a well-established experimental model to mimic human pulmonary allergic inflammation. Several articles have been published using this experimental model to test potential molecules to control the disease as well as to characterize mechanisms related to the type 2 immune response developed in this model. Indeed, on 19 June 2020, in Frontiers in Immunology (Volume 11) and original research (DOI: 10.3389/fimmu.2020.01224) by Wu and colleagues described an important issue related to Prostaglandins E2 inhibiting the IgE production in OVA-induced asthma. Therefore, we believe that this model is appropriate to demonstrate the effect of carvone. However, we agreed that there are others that mimic human pulmonary allergic inflammation.  Thus, we have included a new paragraph mentioning both the relevance and downsides of the OVA model as follows:

Pages 392-399: “The OVA-induced airway allergic inflammation in mice is a well-established experimental model to investigate the effectiveness of new drug candidates for the treatments of allergic asthma [7]. However, recent research by Wu and colleagues [48] raised important issues related to inhibition of IgE production by PGE2 in OVA-induced asthma. Therefore, although we believe that this model is appropriate to demonstrate the differential effects of carvone enantiomers on IgE-mediated allergic inflammation, further research using human-relevant allergens like house dust mite should be conducted to better characterize the therapeutic potential of R-carvone as an anti-asthmatic compound.”

Point 7: Immunomodulatory effects: The group concludes that Carvones affect the adaptive immune response via induction of regulatory IL-10 producing cells (R-Carvone) or a shift to a Th1 response (S-Carvone). Cytokine measurements in BAL were the only experiments performed to substantiate the hypothesis of the effect mechanism of the Carvones. Further experiments are mandatory to prove the effect mechanism in general and later on in detail.

Are the number of Tregs different in lung and draining lymph node?

Are there differences in the local (lung/draining lymph node) response vs the systemic immune response (spleen).

Composition of immune cells.

Cytokine production upon allergen restimulation.

Response 7: We agree that further experiments are needed to characterize the immunoregulatory mechanism by carvone. As previously mentioned, we have not analyzed T cells in the lung and the analysis in draining lymph nodes, as well as in the spleen are relevant and, therefore, will be considered in further studies. This pioneering work points to strong evidence of the differential effects of R- and S- carvone enantiomers on allergic airway inflammation. Nevertheless, it opens many questions to be answered in further research. Thus, aware of the limitations of this study with regard to the characterization of the immunoregulatory mechanisms, in this revised manuscript, we discussed better the potential cell sources of IL-10, as well as the possible impacts of R-carvone-mediated IL-10 inhibition on th2 mediated allergic inflammation (Please, see the changes made in lines 321-346) . We have also tempered our conclusions and a new graphical abstract (Fig. 7) was designed in order to suggest potential action mechanisms by R- and S- carvone considering the findings of our study.

Point 8: The group observes a reduction of eosinophils in BAL upon treatment with Carvones during challenge and sensitization phase. Surprisingly, a reduction of immunoglobulins was only observable following treatment during challenge phase.

How do you explain the anti-inflammatory effectiveness of treatment during both phases.

Were lung tissue inflammation and mucus production also suppressed following treatment in the sensitization phase.

Since, systemic antibodies are induced during sensitization phase, how do you explain the effectiveness of Carvones only during treatment in the challenge phase

Response 8: Thank you for your observation. Since a more complete characterization of the anti-allergic effects of the carvone enantiomers was performed in animals treated during the challenge, we chose two significant parameter (eosinophil recruitment and IgE production) to answer whether the enantiomers would demonstrate anti-allergic effect when the treatment was carried out before sensitization and for this, we do not perform lung analyzes. It remains unknown how the treatment with these enantiomers before each sensitization reduced the number of eosinophils without changing the levels of IgE. Studies indicate that carvone metabolism in humans occurs around 2.4 h. However, the pharmacokinetic profile of carvone enantiomers in mice remain to be investigated. Nevertheless, evidence has suggested that carvone metabolism is likely to be different in humans and rats, as due to enterohepatic, rats can be more sensitive than humans for terpenes. Therefore, studies are needed to elucidate the half-life or carvone enantiomers in mice, as well as the existence of active metabolites and its immunomodulatory effects in the long-term, including with regard to the modulation of IgE production. Despite being produced marjoritly during the sensitization phase, evidence indicates that some additional IgE production may occur due to exposure to OVA during the challenge period (Hurst et al. Modulation of Inhaled antigen-induced IgE Tolerance by Ongoing Th2 Responses in the Lung. The Journal of Immunology April 15, 2001, 166 (8) 4922-4930; DOI: 10.4049/jimmunol.166.8.4922), which in turn could be modulated by the action of mediators such as IL- 10, which would justify the weak effect of R-carvone in reducing IgE titers in mice treated during the challenge phase. This result was better discussed in the revised manuscript (lines 379-386).

Point 9: A little bit more information of the S- and R-Carvones from Sigma should be provided.

How are the preparations of both Carvones performed, are they similar?

The Carvones are diluted in 5% Tween 80 and further diluted in PBS. Are the dilutions comparable? Was diluted Tween 80 tested as a control?

Response 9: The preparation of both substances was similar. They also present comparable dilutions. The control groups received PBS containing 5% tween 80 (vehicle). This is now better explained in the 4.1 section (lines 438-448).

Point 10: In the discussion a differentiated regulation of MMP-2 was presented as an explanation why eosinophils were reduced upon both treatments in BAL but only R-Carvones led to a reduction of inflammatory cells in lung tissue.

A demonstration of a differential regulation of MMP-2 would be beneficial for the manuscript.

Response 10: We agree with you. However, this will be investigated in the future and used in a paper demonstrating the mechanisms associated with the accumulation of inflammatory cells in the lung of S-carvone-treated mice. In this context, our group is now dedicated to investigating the participation of different T cell populations, as well as the involvement of different MMP families on the airway inflammatory profile of allergic mice undergoing S-carvone treatment. Nevertheless, Therefore, we think that it is important to discuss the possibility that  S-carvone could be inhibiting the activity or expression of MMP-2 or other molecules involved in the elimination of inflammatory cells through the lung parenchyma, since this was previously demonstrated for borneol, a monoterpene with a structure similar to that of carvone (30. Dai, J.P.; Chen, J.; Bei, Y.F.; Han, B. X.; Wang, S. Influence of borneol on primary mice oral fibroblasts: a penetration enhancer may be used in oral submucous fibrosis. J Oral Pathol Med 2009, 38(3), 276-281).

Point 11: Minor comments

Please check citations, some citations are missing (46 and further), some citations seem to be wrong.

Please indicate magnification of histological pictures (Fig.2 and Fig.3)

As mentioned before, please ad grades, concentrations and application amounts OVA and all Carvone treatments.

Response 11: Thank You. The citations were checked.

The magnification of histological pictures is described is both materials and methods and in the picture legends.

As previously mentioned, grades, concentrations, and application amounts of OVA and all Carvone treatments were described in the revised manuscript.

Reviewer 2 Report

The study was aimed to investigate the anti-allergic activity of carvone enantiomers in an ovalbumin induced airway allergic inflammation. Although the topic could be of interest for potential readers I found in the texts several major and minor issues that need to be addressed:

Major:

  • Author wrote in the Abstract; “In conclusion, unlike S-carvone, R-carvone has the potential to be used in antiasthmatic drug development.” But why they used carvone only before sensitization and before challenge. If it is to be considered as potential therapeutic in asthma it should be tested above all after Ova challenge and after inducing asthma attack. It is hard to imagine that asthmatics will take carvone or other drugs preventively.
  • Results: Author counted eosinophils in BAL. What about leukocytes? If they performed BAL, they should also count the number of leukocytes and show the data, instead of relying on “data not shown” all the time.
  • 2.2. Authors show leukocyte infiltration into the lung tissue by histology, however what about quantitative data, please use some scale to count the leukocyte layers around blood vessels and bronchi.
  • How they interpret the results on IL-13. They reported reduced eosinophil influx in BAL in all orally treated groups but the level of IL-13 was not reduced. There seems to be effect for Dexamethasone but not significant. The same with mucus production, it was inhibited in R-carvone and DEX groups but it did not correlate with reduction in IL-13 level. Please explain and discuss this.
  • 4.8. Statistical Analyses. Authors wrote that data are presented as means ± Standard Error of Mean (SEM), while in caption to Figures 1, 4 , 5 they state that data are means ± SD, please decide what you present and correct this.

Minor:

  • Author should decide and write some of the words in unified version: In abstract there is “antiallergic” and anti-allergic”
  • Shortcut “BAL” appearing for the first time in abstract should be explained.
  • Results 2.1: “Fig.1B” or “Figure 1B”- please unify
  • Please check carefully the References numbers. The numbering is confusing. Holt et al in the text (line 395) is numbered [46], while in the list is [44]. There is also lack in the list positions no. : 46, 47.

Author Response

Point 1: Author wrote in the Abstract; “In conclusion, unlike S-carvone, R-carvone has the potential to be used in antiasthmatic drug development.” But why they used carvone only before sensitization and before challenge. If it is to be considered as potential therapeutic in asthma it should be tested above all after Ova challenge and after inducing asthma attack. It is hard to imagine that asthmatics will take carvone or other drugs preventively.

Response 1: Thank you for your contribution. This pioneering work points to strong evidence of the differential effects of R- and S- carvone enantiomers on allergic airway inflammation. Nevertheless, it opens many questions to be answered in further research. Considering drug development as a multiphase process, we believe that R-carvone has the potential to be used in the early/pre-clinical stages of anti-asthmatic drug development. In the revised manuscript we discuss the importance of conducting additional studies to evaluate the effectiveness of R-carvone treatment at different stages of the allergic cascade triggered by sensitization and challenge with OVA, as well as of investigating the effect of the post-treatment (after the allergic challenge) with this compound since it more accurately simulates the treatment in asthmatic individuals (lines 372-378).

Point 2: Results: Author counted eosinophils in BAL. What about leukocytes? If they performed BAL, they should also count the number of leukocytes and show the data, instead of relying on “data not shown” all the time.

Response 2: Thank you for your suggestion. In our revised manuscript we provide a more detailed cellular composition of BAL. The number of total leukocytes (Fig 2A), eosinophils (Fig. 2B), mononuclear cells (Fig. 2C), and neutrophils (Fig. 2D) were included. The description of these new data was discussed in materials and methods, results, and discussion.

Point 3: 2.2. Authors show leukocyte infiltration into the lung tissue by histology, however what about quantitative data, please use some scale to count the leukocyte layers around blood vessels and bronchi.

Response 3: Thank you for your suggestion. We totally agree that quantitative data improve the estimation of the effectiveness of Carvone treatment. Despite limited access to research laboratories due to the severity of the COVID-19 pandemic in Brazil, we analyzed the slides and made inflammatory and mucus production scores. These data were inserted in the results (Table 1, Table 2, and figure 3B) and its repercussion for the interpretation of the effects of the enantiomers was discussed in the revised manuscript.

Point 4: How they interpret the results on IL-13. They reported reduced eosinophil influx in BAL in all orally treated groups but the level of IL-13 was not reduced. There seems to be effect for Dexamethasone but not significant. The same with mucus production, it was inhibited in R-carvone and DEX groups but it did not correlate with reduction in IL-13 level. Please explain and discuss this.

Response 4: Since the R-carvone treatment exerted no inhibitory effect on IL-13 production, it was hypothesized the reduced mucus production observed in the lung of the mice treated with this monoterpene resulted from the inhibition of eosinophilic inflammation, although the interference of this terpene on eosinophil activation and mediator secretion needs to be further investigated. However, this terpene was found to stimulate the production of IL-10, a cytokine with key roles in immune regulation during asthma development. Evidence has suggested that, due to its immunosuppressive and anti-inflammatory properties, this cytokine plays a beneficial role in the pathogenesis of asthma. Additionally, studies have demonstrated that CD4+ Th2 cells, which play critical roles in asthma pathogenesis, are directly regulated by IL-10 during allergic airway inflammation. Therefore, the increased IL-10 concentrations found in the BAL of R-carvone-treated mice could justify, at least partially, the reduction in airway inflammatory parameters observed in these animals, including mucus production. These effects seem to be mediated by mechanisms quite different from those underlying the effects of dexamethasone, which significantly inhibited IL-13 production, but had no significant effects on IL-10 concentrations. The possibility that carvone could be modulating other mediators with crucial roles in mucus production, such as Cysteinyl leukotrienes cannot be ruled out and will be further investigated.

In our revised manuscript we have made a significant effort to improve the discussion of these results, as seen on page 10.

Point 5: 4.8. Statistical Analyses. Authors wrote that data are presented as means ± Standard Error of Mean (SEM), while in caption to Figures 1, 4 , 5 they state that data are means ± SD, please decide what you present and correct this.

Response 5: Thank you for the observation. Data are presented as means ± Standard Error of Mean (SEM). We corrected it in the legends of the figures.

Point 6: Author should decide and write some of the words in unified version: In abstract there is “antiallergic” and anti-allergic”

Response 6: We have carefully revised the manuscript and words such as “anti-allergic” and “anti-asthmatic” were written in a unified version.

Point 7: Shortcut “BAL” appearing for the first time in the abstract should be explained.

Response 7: “BAL” is now defined the first time it appears both in the abstract.

Point 8: Results 2.1: “Fig.1B” or “Figure 1B”- please unify

Response 8: Thank you. These terms were unified.

Point 9: Please check carefully the References numbers. The numbering is confusing. Holt et al in the text (line 395) is numbered [46], while in the list is [44]. There is also lack in the list positions no. : 46, 47.

Response 9: Thank you. The references were carefully checked.

Round 2

Reviewer 1 Report

The authors improved several but not all of my mentioned critic points.

Unfortunately, serious points were not answered or a possible answer was postponed on a follow up manuscript.

The manuscript is still very descriptive. It demonstrated a reduced number of eosinophils in the BAL of S and R Carvone treated animals. Surprisingly, reduced inflammation in lung tissue and less goblet cells were only observeable in R-Carvone treated animals. Additionally, increased concentrations of IL-10 in BALF of R-Carvone and increased concentrations of IFN-g in S-Carvone treated animals were detectable. Finally, it demonstrated different effects on OVA specific IgE levels upon treatment with R- and S- Carvones during challenge or sensitization phase.

The group improved quality of the presentation of BAL and lung histology results. Unfortunately, the new presentation raised some minor questions:

  1. Why are only 5 animals instead of 6-8 analyzed?
  2. Why seem the mucus production index and the mucus production scores s different?

Further information concerning the immunological phenotype of lung, draining lymph node or periphery are still completely missing.

The hypothesis of a potential mechanism is toned down but still rest on few descriptive observations.

There is no comment on the missing lung function analysis.

The authors described that in PBS diluted tween 80 was given to the positive and negative control, was there any tween effect? Untreated controls are now missing.

It´s still quite unclear how many animals per experiment were used and how often the experiments were performend. In M&M 6-8 animals are described and that the experiments were repeated twice, in the figure legends >6 are described and in some figures it seems that there are 5 animals analyzed. Please add precise information into the figure legends.

Author Response

Point 1: Why are only 5 animals instead of 6-8 analyzed?

Response 1: Unfortunately, some slides were not in a good quality that would allow an accurate assessment to obtain the score and therefore, they were discarded so as not to compromise the fidelity of the results. Therefore, the new data included in the revised paper were obtained with 5 animals per group. This information is now described in section 4.5.

Point 2: Why seem the mucus production index and the mucus production score different?

Response 2: Although at first glance these results appear different, the pharmacological phenomenon is similar. Note that while the mucus index considers the absolute values of each animal, in the score the animals are classified within a standard with some degree of subjectivity in the pathologist's analysis. Thus, animals within each category of score may show slight differences in mucus index, since the absolute values are detected by the software.

Point 3: Further information concerning the immunological phenotype of lung, draining lymph node or periphery are still completely missing.

Response 3: We agree that determining immunological phenotype of lung, draining lymph node or periphery would improve our paper. However, we have not performed these analyses in the current study. We hope to include them in future research.

Point 4: The hypothesis of a potential mechanism is toned down but still rest on few descriptive observations.

Response 4: The present study is a pioneer in addressing the differential action of the R and S carvone enantiomers. Here, our focus is to demonstrate that differences in the enantiomeric structure of these molecules result in different antiallergic effects. For this, we used a very well- established experimental model and analyzed some of the main cellular and histological parameters, as well as mediators potentially affected in the induction, exacerbation or regulation of eosinophilic inflammation. Thus, we are aware that the findings in this research open questions to be answered in later studies. Still, we believe that our findings bring original data that contribute to the fields of immunopharmacology and preclinical research. In view of the data collected, we seek to raise coherent hypotheses, supported by the literature. Nevertheless, we toned our conclusions, emphasizing that more studies are required to elucidate the molecular mechanisms underlying the effects of each of the molecules investigated in this study. We believe that many of the questions raised will require a multi-step work to be fully responded, since there are several targets to be investigated in this experimental model of asthma, which, although it develops at the pulmonary level, presents significant systemic manifestations.

Point 5: There is no comment on the missing lung function analysis

Response 5: Thank you for your comment. We totally agree that analysis of lung function is very important to understand the effects of the drugs on airway hyper-responsiveness, an important characteristic of asthma. Nevertheless, this complex phenomenon involves a series of events and communication between inflammatory mediators (such as IL-13), immune cells and airway smooth muscle cells. Here, we focused on airway inflammation and its modulation by the carvone enantiomers. Although we believe that the anti-inflammatory activities of carvone could be correlated with an improvement of lung function, it remains to be proven. In the future, we intend to use whole-body plethysmography, ex vivo analyzes in tracheal rings, as well as in vivo analysis in smooth muscle cells to analyze the effects of carvone treatment in the lung functions of allergic mice. We have also revised our conclusions with regard to the anti-asthmatic activity, being more faithful to the proved antiallergic/anti-inflammatory effects.

  1. Paul, J. Zhu How are TH2-type immune responses initiated and amplified Nat. Rev. Immunol., 10 (2010), pp. 225-235

Ribeiro-Filho J, Calheiros AS, Vieira-de-Abreu A, de Carvalho KI, da Silva Mendes D, Melo CB, Martins MA, da Silva Dias C, Piuvezam MR, Bozza PT. Curine inhibits eosinophil activation and airway hyper-responsiveness in a mouse model of allergic asthma. Toxicol Appl Pharmacol. 2013 Nov 15;273(1):19-26.

Point 6: The authors described that in PBS diluted tween 80 was given to the positive and negative control, was there any tween effect? Untreated controls are now missing.

Response 6: PBS diluted in tween 80 was given to the controls as a vehicle, since it was used to dissolve both carvone enantiomers, to ensure that no effect was attributed to the presence or lack of this substance in a specific group. Therefore, since all groups received the same concentration/dose of this substance, the effects of each treatment/ stimulation are guaranteed. Moreover, we and other groups have used tween 80 (5%) to prepare solutions of poorly soluble extracts and isolated compounds and we have noted no intrinsic effect in this model, in comparison with other vehicles such as water, saline and pbs. Thus, animals treated only with the vehicle (PBS/5%) can be considered as untreated, due to the lack of an active compound.

Point 7: It´s still quite unclear how many animals per experiment were used and how often the experiments were performend. In M&M 6-8 animals are described and that the experiments were repeated twice, in the figure legends >6 are described, and, in some figures, it seems that there are 5 animals analyzed. Please add precise information into the figure legends.

Response 7: All experiments were performed twice. Due to loss of sample, some variation in the number of animals may have occurred. However, this was now clearly mentioned in the legends.

Reviewer 2 Report

The manuscript has been much improved. Therefore I recommend it for publication with minor changes:

  • Shortcut BAL is present earlier in the text of abstract, in line 33. So, please explain this abbreviation in this localization, not in line 36.
  • From Fig. 1 delete picture B, since eosinophil number is presented and repeated in Fig. 2 B.

Author Response

Point 1: Shortcut BAL is present earlier in the text of abstract, in line 33. So, please explain this abbreviation in this localization, not in line 36.

Response 1: Thank for your observation. We have corrected this.

Point 2: From Fig. 1 delete picture B, since eosinophil number is presented and repeated in Fig. 2 B.

Response 2: Figure 1 now has only the carvone structure. The number of eosinophils is shown only in figure 2B.